# Co-Stimulatory Molecules during Immune Control of Epstein Barr Virus Infection

**DOI:** 10.3390/biom12010038

**Published:** 2021-12-28

**Authors:** Christian Münz

**Affiliations:** Department of Viral Immunobiology, Institute of Experimental Immunology, University of Zürich, 8057 Zurich, Switzerland; christian.muenz@uzh.ch; Tel.: +41-44-635-3716

**Keywords:** T cells, natural killer cells, cytotoxic lymphocytes, CD27, NKG2D, 4-1BB, PD-1, CTLA-4, 2B4, CD2

## Abstract

The Epstein Barr virus (EBV) is one of the prominent human tumor viruses, and it is efficiently immune-controlled in most virus carriers. Cytotoxic lymphocytes strongly expand during symptomatic primary EBV infection and in preclinical in vivo models of this tumor virus infection. In these models and patients with primary immunodeficiencies, antibody blockade or deficiencies in certain molecular pathways lead to EBV-associated pathologies. In addition to T, NK, and NKT cell development, as well as their cytotoxic machinery, a set of co-stimulatory and co-inhibitory molecules was found to be required for EBV-specific immune control. The role of CD27/CD70, 4-1BB, SLAMs, NKG2D, CD16A/CD2, CTLA-4, and PD-1 will be discussed in this review. Some of these have just been recently identified as crucial for EBV-specific immune control, and for others, their important functions during protection were characterized in in vivo models of EBV infection and its immune control. These insights into the phenotype of cytotoxic lymphocytes that mediate the near-perfect immune control of EBV-associated malignancies might also guide immunotherapies against other tumors in the future.

## 1. Introduction on EBV Infection and Its Malignancies

The Epstein Barr virus (EBV) is a class I WHO carcinogen [1]. At the same time, it is present as a persistent infection in more than 90% of the adult human population [2,3]. Therefore, it is not only one of the most growth-transforming pathogens, readily immortalizing human B cells to lymphoblastoid cell lines (LCLs) in vitro, but also one of the most widely distributed viruses in humans, to which most likely the human immune system had to adapt to prevent EBV-associated malignancies prior to reproductive age. EBV is thought to be primarily transmitted via saliva, e.g., during kissing. This leads to two thirds of European and US populations being infected before the age of two years, and African populations being nearly uniformly infected at this age [4]. EBV crosses the oropharyngeal mucosa most likely by transcytosis [5,6] to infect human B cells in submucosal secondary lymphoid tissues, such as the tonsils [2]. It then drives these infected B cells into proliferation and differentiation through the germinal center reaction to gain access to long-lived memory B cells for persistence. From this memory B-cell compartment, it can reactivate to produce infectious viral particles most likely after B-cell receptor engagement of the respective memory B cells. The immediate early transcription factor BZLF1 of EBV drives this lytic reactivation. The resulting viral particles can then infect epithelial cells from the basolateral side, presumably for another round of lytic replication before being shed into the saliva for transmission. B-cell proliferation and differentiation is supported by so-called latency gene transcription programs. In naïve B cells, latency III is found with the expression of all six nuclear antigens (EBNA1, 2, 3A-C, and -LP), two latent membrane proteins (LMP1 and 2), two EBV-encoded small RNAs (EBER1 and 2), and more than 40 miRNAs of the BHRF1 and BART clusters [7,8,9]. In germinal center B cells then, only EBNA1, LMP1, LMP2, EBER, and BART miRNA expression can be found [8]. This is called the latency IIa program. Finally, in memory B cells, only non-translated EBER and BART RNAs are expressed (latency 0) with transient expression of EBNA1 during homeostatic proliferation (latency I) [10,11]. Accordingly, these latent gene products fulfill important functions along this B-cell differentiation trajectory. EBNA2 drives early proliferation and EBNA3A/3C prevent apoptosis during this early proliferation phase [3]. LMP1 and 2 provide proliferation and survival signals for germinal center B cells [12]. EBNA1 maintains the viral DNA in all proliferating cells [13]. These latent EBV gene product functions are also abused by EBV-associated malignancies [14]. Post-transplant lymphoproliferative disease (PTLD) in transplant patients and immunoblastic lymphomas in HIV-infected individuals often express latency III, while EBV-associated Hodgkin’s lymphoma expresses latency IIa and Burkitt’s lymphoma latency I. In addition, EBV is also present in around 10% of gastric carcinoma (latency I) and nasopharyngeal carcinoma (latency II) which constitute the bulk of EBV-associated tumors that make up 1–2% of all human cancers [15]. Thus, the latency programs that can be found in EBV-associated malignancies include expression of viral oncogenes that are present in healthy virus carriers, and the human immune responses needs to keep these constantly in check during viral persistent infection.

## 2. T Cells as the Cornerstone of EBV-Specific Immune Control

This immune control is primarily mediated by cytotoxic lymphocytes, including natural killer (NK), NKT, and γδ T cells, but primarily αβ T cells, especially CD8^+^ T cells [16,17,18,19]. Several lines of evidence point in this direction, namely expansion of these lymphocyte compartments during symptomatic primary infection, such as infectious mononucleosis (IM), their efficacy in controlling EBV infection in preclinical animal models, and their genetic, co-infection-induced or iatrogenic defects in patients that suffer from EBV-associated pathologies. During IM NK, NKT, γδ T, and CD8^+^ αβ, T cells expand [20,21,22]. Early differentiated NK cells primarily target lytic EBV infection [21,23]. NKT cells might preferentially target latency II infected B cells [24,25]. Furthermore, Vγ9Vδ2 T cells get primarily stimulated by latency I infected B cells [22,26]. αβ CD8^+^ T cells that expand during IM are primarily directed against lytic EBV antigens [27,28], but adoptive transfer of T-cell lines specific for EBNA1 or LMP1 and 2 can cure EBV-associated malignancies in patients [29,30,31,32,33]. Therefore, IM seems to drive the expansion of several innate and adaptive lymphocyte compartments, suggesting their involvement in the immune control of EBV.

Their contribution to EBV-specific immune control can be tested in preclinical in vivo models, such as non-human primates and mice with reconstituted human immune system components (humanized mice) [34,35]. Accordingly, NK cell depletion leads to higher viral loads and increased lymphomagenesis in EBV-infected humanized mice [36,37]. This only affects infection with wild-type, but not lytic infection deficient BZLF1 knock-out EBV [36]. Moreover, mixed HLA disparate human immune reconstitution improves EBV-specific immune control, presumably by limiting inhibitory NK cell receptor engagement by EBV-infected B cells of the co-reconstituted donor [37]. Moreover, adoptive transfer of CD8^+^ NKT cells that expand during EBV infection limits lymphomagenesis in humanized mice [25]. Similarly, expansion or adoptive transfer of Vγ9Vδ2 T cells in humanized mice limited PTLD-like disease [38,39]. In addition to these innate lymphocytes, αβ CD8^+^ T cells massively expand during EBV infection of humanized mice [40,41,42,43]. Furthermore, their antibody-mediated depletion increases viral loads and EBV-associated lymphomagenesis [40,43,44,45,46]. The role of CD4^+^ T cells in EBV-specific immune control in humanized mice is less clear. Their depletion by antibodies or HIV co-infection increases viral loads and lymphomagenesis [40,45]. Furthermore, immune-suppressive FK506 (tacrolimus) treatment mainly affects IL-2 production and activation of CD4^+^ T cells in humanized mice and also increases viral loads and lymphomagenesis [47]. Finally, adoptive transfer of cytotoxic EBV-specific CD4^+^ T cells can control LCL growth in humanized mice [48]. However, non-cytotoxic CD4^+^ T-cell functions can also support EBV-associated lymphoma growth and possibly latency I/II infection in humanized mice [49,50]. Thus, depending on their polarization, CD4^+^ T cells might play pro- or anti-viral roles during EBV infection.

However, their depletion by HIV co-infection also increases EBV-associated lymphomagenesis in patients [51]; even so, MHC class II deficient patients with a compromised CD4^+^ T-cell compartment usually have no problems with persistent EBV infection [52]. Indeed, the notion that lymphocyte cytotoxicity-supporting functions of CD4^+^ T cells are important for EBV-specific immune control, while other helper cytokines might rather drive immune pathology or even lymphomagenesis, is supported by patients with primary immunodeficiencies due to mutations in the cytotoxic machinery [53,54,55]. These often suffer from cytokine-driven hemophagocytic lymphohistiocytosis (HLH), which is driven by poorly controlled EBV infection in the absence of cytotoxicity. The importance of cytotoxic lymphocytes in the immune control of EBV is also supported by other primary immunodeficiencies that predispose for EBV-associated pathologies. These include factors for their development (GATA2, MCM4) and others that are required for the expansion of these lymphocytes (XIAP, STK4, CTPS1) [16,17,18,19]. Moreover, the importance of T cells is underscored by the susceptibility of individuals with mutations in molecules of T-cell receptor signaling (ITK, RASGRP1, ZAP70, PI3K). However, while these factors are generally required for human T-cell development and function, mutations in co-stimulatory molecules and their mechanistic studies provide insights into the particular interactions of T cells with EBV-infected cells and/or antigen-presenting cells that are required for the control of this ubiquitous human tumor virus. The discussion of these interactions is the main focus of this review.

## 3. Co-Stimulatory CD27/CD70 Signaling

Homozygous mutations in the tumor necrosis factor (TNF) receptor superfamily member CD27 or its ligand CD70, which have so far been recorded in 49 patients from 29 families, predispose affected individuals to EBV-induced lymphoproliferations, often developing into lymphomas such as Hodgkin’s lymphoma [19,56]. Most of these patients benefit from B-cell depletion by antibody therapy or bone marrow transplantation as curative treatments [56]. Several, but not all, of their EBV-specific T-cell responses are reduced and demonstrate some deficiencies in killing EBV-transformed B cells [56,57,58]. It was postulated that this functional deficiency could result from diminished expansion and differentiation of EBV-specific T cells in the absence of CD27 that would result in their decreased surface expression of co-stimulatory molecules, such as 2B4 and NKG2D, which are directly involved in recognizing EBV-infected B cells [19]. 

The CD27 interaction with CD70 is in addition to the CD28 interaction with CD80 and CD86, considered to be one of the key co-stimulations during the initiation of T-cell responses [59,60]. Therefore, it is surprising that CD27 deficiency predominantly predisposes for EBV-associated pathologies and not for susceptibilities to most other infectious diseases, and that CD27 or CD70 deficient patients still have some EBV-specific T-cell responses. Along these lines, it was recently found in humanized mice that antibody-mediated blockade of CD27 primarily limits expansion of lytic EBV antigen-specific CD8^+^ T-cell responses with intact expansion of latent EBV antigen-specific CD8^+^ T-cell responses [61]. In addition, CD27 blockade also inhibited cytotoxicity of lytic EBV antigen-specific CD8^+^ T-cell clones against EBV-transformed autologous LCLs (Figure 1), and CD70 is upregulated upon B-cell transformation by EBV. Accordingly, CD27 blockade increased viral load and EBV-infected B-cell lymphoproliferations during wild-type but not BZLF1 deficient lytic replication incompetent EBV infection. Therefore, CD27 might play a role during expansion and EBV-transformed B-cell recognition for a subset of EBV-specific T-cell responses that are required for EBV-specific immune control. This might explain the emergence of EBV-associated pathologies despite the presence of some EBV-specific T-cell responses in patients with homozygous CD27 or CD70 deficiencies.

## 4. Co-Stimulation by 4-1BB

Another TNF receptor superfamily member that is required for EBV-specific immune control is CD137/4-1BB [62,63,64,65]. Less than 10 patients have been so far diagnosed with this deficiency and suffer from lymphoproliferations and lymphomas. However, one of these EBV infections spreads to T cells (Figure 1), possibly assisted by a concomitant loss of function mutation in the p110δ subunit of phosphatidylinositol 3-kinase (PIK3CD) [64]. Furthermore, in two studies, siblings with the same recessive loss-of-function mutations in 4-1BB were identified that did not present with any clinical symptoms [63,64]. Therefore, 4-1BB does not seem to be as essential for EBV-specific immune control as CD27. Nevertheless, patients with 4-1BB mutations that suffer from EBV-driven lymphoproliferations can be successfully treated with B-cell depleting therapy [62,63,65] if infection has not spread to T cells [64] and EBV-associated lymphomas have not lost the target antigen CD20 [65]. Alternatively, bone marrow transplantation offers a curative treatment [62,65].

Similar to CD27 deficiency, T-cell expansion and cytotoxicity of EBV-infected cells is compromised by 4-1BB deficiency (Figure 1) [62,63]. This can also be mimicked by antibody-mediated 4-1BB blocking in vitro [62]. Accordingly, 4-1BBL is upregulated by EBV transformation of B cells [66]. The lower penetrance of EBV-associated pathologies in individuals with homozygous 4-1BB mutations, as demonstrated by healthy siblings of the patients that carry the same mutations, suggests that compensatory co-stimulation might rescue EBV-specific immune control in these siblings. Along these lines, CD28 stimulation can compensate for 4-1BB deficiency for T-cell proliferation [63]. Thus, 4-1BB seems to support T-cell expansion and cytotoxicity against EBV-infected cells but is not as strictly required for these functions as CD27 during EBV-specific immune control. 

## 5. Co-Stimulatory Molecules of the SLAM Receptor Family

Loss of function mutations in the associated protein of signaling lymphocyte activation molecules (SLAM), called SAP and encoded by the SH2D1A gene, cause x-linked lymphoproliferative disease type 1 (XLP1) [67,68,69,70,71]. The majority of the affected individuals develop viremia and HLH after primary EBV infection, and up to 50% develop lymphomas that are associated with this γ-herpesvirus, while other viral infections, including α- and β-herpesviruses, are controlled by these patients [72,73,74]. Bone marrow transplantation and antibody-mediated B-cell depletion are successful treatments in XLP1 patients [73,75,76,77]. Thus, SAP deficiency renders affected individuals exquisitely sensitive to EBV-associated pathologies and thereby implicates SLAM receptors in EBV-specific immune control.

SAP converts SLAM signaling from inhibitory to activating [78,79]. Accordingly, cytotoxic lymphocytes such as CD8^+^ T cells and NK cells cannot efficiently kill EBV-transformed LCLs without SAP (Figure 1) [80,81]. In addition, NKT cells are lacking in XLP1 patients [82]. The inability of cytotoxic XLP1 lymphocytes to target EBV-transformed B cells is at least in part due to the inability of the two SLAM receptors 2B4 and NTB-A to activate CD8^+^ T cells and NK cells in this interaction without SAP [83,84]. These two receptors are in particular required for cytotoxic immune synapse formation with B cells, but not other antigen-presenting cells, such as dendritic cells (DCs) and fibroblasts [80,85,86]. The 2B4 ligand CD48, especially, is also strongly upregulated upon EBV infection of B cells [66]. This strict requirement for SAP in cytotoxic lymphocyte recognition of EBV-transformed B cells results in EBV-specific CD8^+^ T cells of female carriers of SAP mutations expressing uniformly wild-type SAP, while CD8^+^ T cells against human cytomegalovirus or influenza virus express mutant or wild-type SAP [86]. Furthermore, EBV-specific CD8^+^ T cells with revertant SAP mutations can be found in XLP1 patients [87]. The lack of eliminating EBV-transformed B cells in the absence SAP can at least in part be mimicked by antibody blocking of 2B4 [44]. This leads to higher viral loads and lymphoma formation in humanized mice, while CD8^+^ T cell expansion upon EBV infection is not altered by 2B4 blockade in this model. Furthermore, simultaneous antibody-mediated depletion of CD8^+^ T cells leads to loss of EBV-specific immune control that cannot be worsened by 2B4 blockade. This suggests that 2B4 is mainly required on CD8^+^ T cells for eliminating EBV-infected B cells. Thus, the SLAM receptors, especially 2B4, seem to be mainly required for lymphocyte cytotoxicity against EBV-infected B cells but much less for EBV-specific T-cell expansion. 

## 6. Stimulatory NKG2D Engagement

Another receptor that is engaged to kill EBV-infected B cells is NKG2D [23,88]. NKG2D ligands seem to be mainly upregulated during the transition from latently EBV-infected B cells to viral lytic replication (Figure 1) [23]. This is also consistent with preferential recognition of B cells with lytic EBV infection by NK cells which uniformly express NKG2D [21,23,36]. Accordingly, NK cells limit wild-type EBV infection and associated lymphomagenesis in humanized mice, but do not influence infection with BZLF1-deficient lytic EBV infection incompetent virus [36]. In the primary X-linked immunodeficiency with magnesium defect, the Epstein-Barr virus infection, and neoplasia (XMEN), magnesium transporter 1 (MAGT1) is mutated and the diminished intracellular magnesium levels compromise glycosylation and surface expression of NKG2D [88]. Restoring NKG2D surface expression by magnesium supplementation was able to increase NK cell cytotoxicity against EBV-transformed LCLs. Thus, NKG2D, in addition to SLAM receptors, seems to be required for the direct recognition of EBV-infected B cells. However, NKG2D-mediated recognition might primarily be important for the immune control of lytic EBV infection.

## 7. Modifying CD2 Co-Stimulation by CD16

The low affinity IgG receptor CD16 (FcγRIII) was found to be mutated or deleted in a few patients that suffer from EBV-associated pathologies [89,90,91]. In a family with three siblings that all lacked CD16A on NK cells but retained CD16B on monocytes, elevated EBV viral loads were associated with CD16A deficiency [91]. In one of these, persistent EBV viremia with clinical symptoms was associated with T-cell infection by EBV (Figure 1). CD16A with one altered amino acid was also the cause of poorly controlled herpesvirus infections in the originally described two patients [89,90]. How the low affinity FcγRIIIA is required for EBV-specific immune control is unclear, however, because antibody responses do not seem to be required to keep EBV in check [18]. It has been proposed that it either influences the function of the adhesion molecule CD2 that binds to CD58 on target cells [90], or maturation of NK cells to high CD2 expression [91]. KSHV co-infection also diminishes CD2 expressing NK cells and increases EBV-associated lymphomagenesis in humanized mice [92,93]. In one of the original studies that identified poorly controlled EBV infection in patients with CD16A mutations, direct association of CD16A with CD2 was demonstrated, thereby modifying NK cell cytotoxicity (Figure 1) [90]. Furthermore, CD16A mutant NK cells were less capable of forming this association with CD2 and compromised in their CD2-dependent cytotoxicity. CD58, the ligand of CD2, is transiently upregulated in the first five days after EBV infection of B cells [66]. This period is dominated by EBNA2-driven proliferation of infected B cells without LMP expression, the so-called latency IIb program [94]. Thus, mutant CD16A or CD16A deficiency is associated with loss of EBV-specific immune control but possibly not by modifying IgG opsonized target cell recognition, and rather by loss of cytotoxic immune synapse formation due to decreased CD2 levels or function. 

## 8. The Co-Inhibitory Receptors PD-1 and CTLA-4

Not only the loss of co-stimulatory receptors, but also the loss of co-inhibitory receptors was found to cause EBV pathologies. Among these, CTLA-4 deficiency, often even haploinsufficiency, is associated with increased viral loads in more than half of the individuals, as well as EBV-associated lymphoma and gastric carcinoma formation [95,96,97]. Diffuse large B-cell lymphomas, Burkitt lymphomas, and Hodgkin lymphomas have been observed in CTLA-4-deficient patients [96]. By now, more than 150 CTLA-4-insufficient patients have been described, and they present usually with decreased T-cell but increased regulatory T-cell numbers (Figure 2) [95]. Accordingly, both effector T-cell exhaustion and/or increased T-cell regulation could account for the lack of EBV-specific immune control. The increased T-cell lymphoproliferation causing activation induced cell death, which might be the cause of T-cell lymphopenia in these patients, can be treated with recombinant CTLA-4-Fc reagents, such as Abatacept and Belatacept [95,97,98]. However, EBV reactivation needs to be monitored during such treatments. Thus, the role of CTLA-4 during EBV-specific immune control is incompletely understood.

Another inhibitory co-receptor on T cells, PD-1, was also explored during EBV infection in humanized mice. Surprisingly, increased viral loads and EBV-associated lymphomagenesis was observed during PD-1-specific antibody blockade (Figure 2) [42,99]. Only in an EBV-infected cord blood transfer model into immune compromised mice did PD-1 blockade alone or together with CTLA-4 blockade decrease EBV-associated lymphomagenesis [100]. In humanized mice, the loss of EBV-specific immune control during PD-1 blockade was associated with elevated levels of the immune-suppressive cytokine IL-10, which is also produced by a subset of regulatory T cells [42,99]. Both PD-1 ligands, PD-L1 and PD-L2, are upregulated on B cells after EBV infection [66]. Along these lines, some cancer patients that receive therapeutic PD-1-targeted immune check-point blockade develop neurological symptoms. In some of these, EBV viral loads were elevated, and both EBV-infected B cells as well as EBV-specific T cells were suggested to home into the central nervous system (CNS), in part being responsible for the observed neurological side effects [101]. Such respective clonal B- and T-cell expansions, pointing to EBV-infected B- and EBV-specific T-cell homing to the CNS, were, however, primarily reported in one patient after immune checkpoint blockade. Therefore, it is tempting to speculate that both CTLA-4 haploinsufficiency and PD-1 blockade might increase regulatory T-cell function during EBV infection, resulting in virus-associated pathologies.

## 9. Conclusions and Outlook

The summarized studies above highlight molecular requirements for the near-perfect immune control of EBV and its associated malignancies by cytotoxic lymphocytes. Similarly, robust cytotoxic T and innate lymphocyte responses would also be desirable to target other infectious diseases or tumors. Thus, identifying the means to elicit such robust cell-mediated immune responses by vaccination or adoptive lymphocyte transfer would be highly desirable. Maybe also for this goal, EBV infection can provide us with some interesting insights.

As discussed, EBV has a near-exclusive tropism for human B cells. Indeed, it remains so far unclear under which physiological circumstances the virus infects other cells, such as epithelial and T cells from which EBV-associated pathologies emerge [14]. This particular tropism of EBV for B cells indicates primarily cross-presentation by DCs from EBV-infected B cells [102] and/or T-cell stimulation by EBV-infected B cells [103] as the source of EBV-specific T-cell responses. Along these lines, EBV-based and DC-targeted vaccine formulations elicited only low T-cell responses in humanized mice [104,105], while EBV-based and B-cell-targeted vaccination induced protective T-cell responses in the same model [106]. Furthermore, B cells activated by transgenic LMP1 expression were able to elicit tumor-associated, antigen-specific CD4^+^ and CD8^+^ T cells in a mouse model [107]. Therefore, the molecular requirements of this B-cell-elicited T-cell stimulation should be characterized in the future. These might provide complementary pathways to DC-mediated T-cell stimulation in eliciting cytotoxic cellular immune control. Thus, multiple antigen-presenting cell populations, including both DCs and B cells, could be targeted to elicit robust T-cell-mediated immune control, not only of EBV infection and associated malignancies, but also for tumor immunology in general.

## Figures and Tables

**Figure 1 biomolecules-12-00038-f001:**
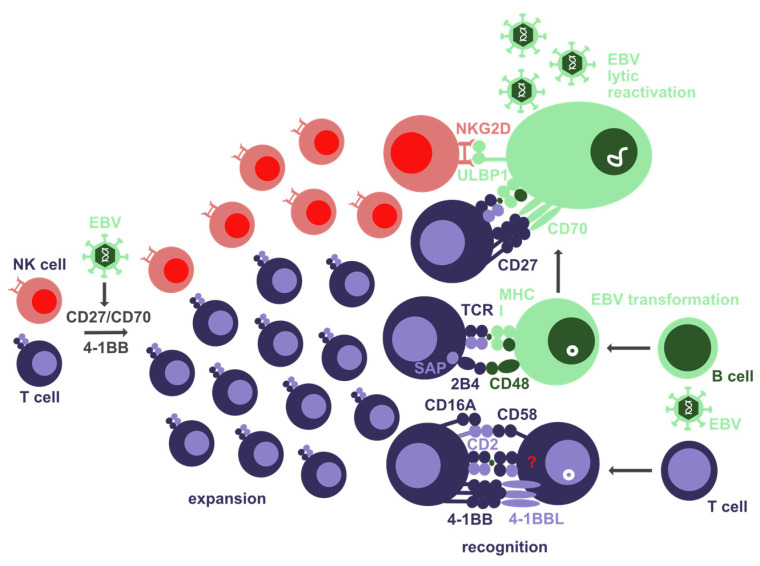
Co-stimulatory molecules that are involved in T and NK cell expansion as well as EBV-infected target cell recognition by these cytotoxic lymphocytes. The interaction of CD27 with CD70 as well as 4-1BB seems to be involved in T-cell expansion after EBV infection. SAP-associated SLAM receptors, such as 2B4, contribute to T-cell recognition of EBV-transformed B cells. CD27 and NKG2D are involved in T and NK cell recognition of lytically EBV-infected B cells, respectively. 4-1BB and CD2 (assisted by CD16A) might be involved in EBV-infected T-cell recognition.

**Figure 2 biomolecules-12-00038-f002:**
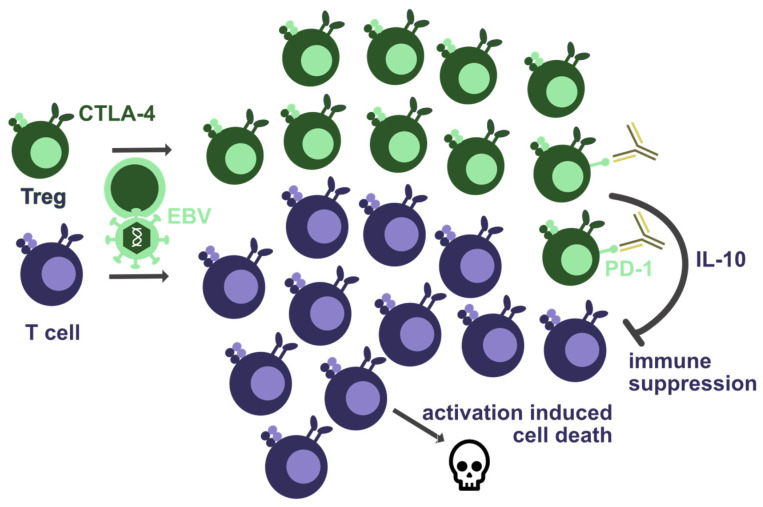
Co-inhibitory receptors that are required for EBV-specific immune control. Haploinsufficiency of CTLA-4 seems to lead to T-cell lymphopenia and increased regulatory T-cell (Treg) numbers. This could result from activation-induced cell death due to T-cell hyperactivation, or from suppression by Tregs due to their diminished CTLA-4-mediated inhibition. Furthermore, PD-1 blockade by antibodies could further increase Treg-mediated suppression of EBV-specific immune control, e.g., via cytokines such as IL-10.

## Data Availability

Not applicable.

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
