# Peer review of "Co-Stimulatory Molecules during Immune Control of Epstein Barr Virus Infection"

_biomolecules, 2021, doi:10.3390/biom12010038_

Round 1
Reviewer 1 Report
Dear author,
the present manuscript is an inteersting review, but where is the novelty of it and why now?
The abstract needs to be re-written, it is very poor, and does not sound like a catching attention to it for the scientific community.
The manuscript has a good structure but lacking in providing detail and relevant information, each section is writting in a "compressing" manner and does not sound scientifcally enough.
Line 242-243: "In one study..." - please re-consider and rewrite such way of refering to previous studies. Please check in the whole manuscript.
Conclusions and outlook is very very kind a summarizing fast some thoughts but needs to be improved this section. What are the future perspectives?
Author Response
I thank both reviewers for their suggestions which have further improved my review. Please find my responses to the individual concerns of the reviewers in italics below. I high-lighted the manuscript changes by underlining in the revised manuscript text.
the present manuscript is an interesting review, but where is the novelty of it and why now?
The novelty lies in a combination of discussing additional co-stimulatory and co-inhibitory molecules being recently identified as crucial for EBV specific immune control (4-1BB and CTLA-4) and the function of some of these being functionally characterized in novel in vivo models of EBV infection and cell mediated immune control (CD27 and PD-1).
The abstract needs to be re-written, it is very poor, and does not sound like a catching attention to it for the scientific community.
I have now modified the abstract to emphasize the above-mentioned novel points in more detail.
The manuscript has a good structure but lacking in providing detail and relevant information, each section is written in a "compressing" manner and does not sound scientifically enough.
I now expanded the discussion in certain sections, especially in the sections with more speculative character on CD16A or PD-1 and CTLA-4 which were high-lighted by both reviewers.
Line 242-243: "In one study..." - please re-consider and rewrite such way of referring to previous studies. Please check in the whole manuscript.
I have now clarified that this finding belongs to one of the original studies in which CD16A mutations were found to be associated with poor EBV specific immune control.
Conclusions and outlook is very very kind a summarizing fast some thoughts but needs to be improved this section. What are the future perspectives?
I have now expanded the conclusions and outlook to include more future perspectives on the characterization of molecular pathways that B cells use to stimulate protective cytotoxic lymphocyte responses against infectious disease agents and cancers.
Reviewer 2 Report
This review on the role of co-stimulatory molecules in immune control of Epstein Barr virus infection is interesting and the approach with using information of patients with mutations in the costimulatory molecules is novel and interesting. The review is complete, the figures are not very easy to comprehend. It took quite a bit of studying to understand the figures and it would be nice if the could be improved. Although I can imagine it is not easy. The part on CTLA4 and PD1 is more based on a mouse model and tumors is it also translatable to the human situation as well? Is the expression of PD-L1 and PD-1 similarly high and do those molecules play an equal role in infection control or more in exhaustion and/or suppression by tumors. Maybe we need more speculation on that part?
Author Response
I thank both reviewers for their suggestions which have further improved my review. Please find my responses to the individual concerns of the reviewers in italics below. I high-lighted the manuscript changes by underlining in the revised manuscript text.
This review on the role of co-stimulatory molecules in immune control of Epstein Barr virus infection is interesting and the approach with using information of patients with mutations in the costimulatory molecules is novel and interesting. The review is complete, the figures are not very easy to comprehend. It took quite a bit of studying to understand the figures and it would be nice if the could be improved. Although I can imagine it is not easy.
I have now further improved the figures by adding additional labels to characterize the different EBV infection stages, distinguish expansion and recognition, as well as clarify that T and NK cell expansion is driven by EBV infection in figure 1, and by high-lighting immune suppression in figure 2.
The part on CTLA4 and PD1 is more based on a mouse model and tumors is it also translatable to the human situation as well? Is the expression of PD-L1 and PD-1 similarly high and do those molecules play an equal role in infection control or more in exhaustion and/or suppression by tumors. Maybe we need more speculation on that part?
I have now expanded the evidence for inhibition by CTLA-4 and PD-1 being required for efficient EBV specific immune control in patients. While the first paragraph on CTLA-4 haploinsufficiency already only discussed observations in patients, I have now clarified that the detailed characterization of one patient after PD-1 immune checkpoint blockade found clonal expansions of EBV infected B cells and EBV specific T cells in the central nervous system. Finally, I have inserted a statement that both PD-1 ligands are up-regulated during B cell transformation by EBV.
Round 2
Reviewer 1 Report
The author adressed all reviwers' comments/suggested. The mansucript can be accepted.